# The Variety of the Stress–strain Response of Silicone Foam after Aging

**DOI:** 10.3390/polym14173606

**Published:** 2022-09-01

**Authors:** Zhaoqun Shao, Min Zhu, Tianxi Liang, Fei Wu, Zijian Xu, Yang Yang, Yilong Liu

**Affiliations:** College of Nuclear Science and Technology, Naval University of Engineering, Wuhan 430033, China

**Keywords:** silicone foam, aging, stress–strain curve, constitutive model

## Abstract

The mechanical properties of silicone foam will degrade when exposed to environmental loads such as temperature and pressure for a long time. In recent years, the variation law of the stress–strain response of silicone foam during the aging process has received more and more attention, but there are few works that quantitatively analyze the variation of the stress–strain response. In this work, we quantitatively analyzed the variation law of the stress–strain response of silicone foam during aging by the constitutive model. Firstly, the accelerated aging test of silicone rubber foam under long-term compressive strain was carried out, and its compression set, stress relaxation and strain stress curves of different aging degrees were obtained. Further, degenerate trajectory equations for the compression set and stress–relaxation were obtained. In addition, the hyper-foam constitutive model was obtained by fitting stress–strain curves, and the changes in the model parameters after aging were studied. The results show that the compressed set and stress–relaxation are exponential functions of time, while different to existing research findings, we found that the stress–strain curves do not change monotonically with increasing time, which first softens, then hardens, and finally softens. Additionally, to better understand the changing trend of the stress–strain response, the correlation between the stress–strain curve and the compression set and stress–relaxation was discussed qualitatively. Finally, in the stage of monotonic change of the stress–strain curve, the exponential function of the model parameters with the increase of aging time was obtained.

## 1. Introduction

Silicone rubber foam is widely used in aerospace and automotive fields due to its excellent shock absorption, thermal insulation, thermal stability, and good chemical stability [1,2,3], while due to environmental stress such as heat, oxygen, mechanical load, radiation, and chemical media [4,5,6], the physical and chemical properties of silicone foam will gradually deteriorate [7,8]. Aging will destroy the molecular structure of polymers by cross-linking or degradation, and the dispersion of additives such as silica in rubber will deteriorate, which will lead to stress–relaxation, permanent deformation, reduced tensile strength, and elongation at break [9,10]. The gradually reduced mechanical properties will shorten the service life and reduce reliability. Therefore, it is very important to study the change law of mechanical properties of silicone foam under typical environmental factors for prolonging product life and improving product reliability.

The degradation of mechanical properties is usually measured by compression set, stress–relaxation, tensile strength, and elongation at break [11,12]. However, in recent years, more and more attention has been paid to the study of the variation law of the stress–strain response of silicone foams during aging. Moreover, existing studies usually study the change of the stress–strain response by establishing constitutive models [13,14,15], which usually use the Mooney–Rivlin or Ogden model to describe the mechanical response of silicone rubber foam. At present, there are many studies on irradiation-related aging constitutive models, and there is a good linear relationship between irradiation dose and cross-linking density, which is conducive to the establishment of related models. However, the most common working environment for silicone foam is long-term compression [16,17,18]. Despite this, little work has been carried out on the constitutive model of silicone foam under long-term compression and thermal working conditions [19]. Lou et al. [19] established a hyper-foam constitutive model of silicone rubber foam under high temperature and compression, which established the model parameters as a function of time and temperature by fitting experimental data. Liu et al. [20] investigated the aging properties of ethylene-propylene-diene monomer (EPDM) rubber under different pre-compressions and used the first-order Mooney–Rivlin constitutive model as the constitutive model of EPDM. Additionally, the relationship between model parameters and pre-compression and time is established by fitting the experimental data. Maiti et al. developed a two network age-aware constitutive model for a 3D printed polymeric foam based on Ogden hyper-foam [21]. Previous studies have shown that silicone foam will harden (or soften) during aging due to the increase (or decrease) of cross-linking density [1,2,3,19,21]. In those research, the stress–strain curve changes monotonically, either hard or soft, which brings convenience to analyze the effect of aging on the stress–strain response of silicone foam [22,23]. However, the actual situation is more complex. The simultaneous existence of oxidative chain breaking and cross-linking during aging leads to the non-monotonic change of stress–strain response [24]. The irreversible failure and buckling deformation of silicone foam during aging also affect its stress–strain response [25]. Therefore, in our work, different from previous reports, the stress–strain response does not change monotonously with the increase of aging time, which softens first and then hardens, and finally softens. This leads to difficulties in establishing aging-related constitutive models. Hence, further investigation is necessary to study the change law of the stress–strain response for aged silicone foam.

In this work, the compression set, stress–relaxation, and a series of stress–strain curves of silica foam under long-term compressive strain were obtained by accelerated aging tests and uniaxial compression tests. Additionally, the degradation paths of the compression set and stress–relaxation after aging for different times were obtained by fitting the experimental data. Furthermore, to analyze the change of stress–strain response with the deepening of aging, the Hyper-foam model was used to fit the test data. By analyzing the variation law of model parameters, in the monotonic change stage of stress–strain curve, the function of constitutive model parameters with aging time was established.

## 2. Materials and Methods

### 2.1. Main Materials and Equipment

The silicone foam rubber used in this article is a type of elevated temperature vulcanized foamed silicone rubber, which was purchased from Xueru seal Co., Ltd., Zhejiang, China (https://www.1688.com/huo/detail-548728181984.html accessed on 20 August 2022) and used as received. The bought sample is a sealing strip, 20 mm wide and 10 mm thick. The silicone foam has a white, adhesive layer with a cell density of 46%. The foams are processed into cube samples with a dimension of 10 mm × 10 mm × 10 mm. To ensure the uniformity of materials, only the middle part is taken during the process. To reduce the error caused by uneven foaming of silicone foam material, the average value of three samples is taken as the test data.

### 2.2. Test Conditions and Procedures

First, fix the sample in the fixture (reference to GB-T 1683), and adjust the height of the limiter to ensure that the silicone foam is under 40% compressive strain. Then, the fixture is placed into an air-circulating oven (Shanghai Test Instrument Factory Co., Ltd., Shanghai, China (https://www.foodjx.com/st196877/) accessed on 20 August 2022) for the accelerated thermal aging test. The samples are aged at 125 °C for a total period of up to 192 h and then removed from the air-circulating oven at specific time points (8, 16, 24, 36, 48, 72, 96, 144, and 192 h). Then, they are placed at 25 °C for 24 h until the elastic deformation is completely restored, and the compression set, stress–relaxation, and strain–stress curve are obtained by a universal testing machine (the loading rate is 10.0 mm/min) and thickness gauge. Note that stress–relaxation is measured by the discontinuous stress–relaxation test method on a universal testing machine. The stress–relaxation curve of silicone foam is shown in Figure 1. It implies that the load applied to the silicone foam changes greatly at the beginning and gradually stabilizes from 600 s to 1000 s. Therefore, this article intercepts the load at 1000 s as the stable load.

## 3. Results and Discussion

### 3.1. Variation of Compression Set, Stress–Relaxation, and Stress–strain Curve

The compression set refers to the irreversible change in the height of silicone foam after aging. Referring to GB T 7759, the material compression set can be expressed as:(1)$$C_{t}=\frac{h_{0}-h_{t}}{h_{0}-h_{s}},\;$$
where $C_{t}$ is the compression set; $h_{0}$ is the initial height before aging; $h_{t}$ is the recovered height after aging; $h_{s}$ is the height of the limiter.

Stress–relaxation refers to the reduction of stress after aging for different times in compression. The equation for calculating stress retention can be expressed as:(2)$$L_{t}=\frac{\sigma_{t}}{\sigma_{0}},\;$$
where $L_{t}$ is the stress retention; $\sigma_{0}$ is the stress before aging; $\sigma_{t}$ is the stress after aging.

Figure 2 show the compression set and stress–relaxation curves of the compressed silicone rubber foam aging at different times. Previous studies show that the compression set and stress–relaxation can be written as an exponential function and logarithmic function of time, respectively [5,12], as shown in Equations (3) and (4).
(3)$$C_{t}=1-A\exp(-k_{c}t),\;$$
(4)$$L_{t}=L_{0}-k_{L}\ln(t),$$
where *A* is a constant; $k_{c}$ and $k_{L}$ are the coefficients related to temperature; $L_{0}$ is the stress retention coefficient of silicone foam before aging, theoretically $L_{0}$ = 1.

By fitting the test data with the least square method, the compression set and stress–relaxation degradation trajectory of silicone foam at 125 °C are obtained, as shown in Equations (5) and (6).
(5)$$\ln(1-C_{t})=-0.0018t-0.1106,\;$$
(6)$$L_{t}=0.9988-0.0906\ln(t).\;$$

From Figure 2 and the degradation trajectory equation of compression set and stress–relaxation, it can be observed that the stress–relaxation changed rapidly during the early stage and then changed slowly, while the permanent compression deformation changed relatively evenly. Previous studies have shown that the stress–strain response is related to stress–relaxation and compression set [21]. It can be concluded that the deterioration of the stress–strain curve caused by aging is coupled and affected by stress–relaxation and the compression set. Assuming that there is no stress–relaxation, the increase of the compression set will lead to the decrease of actual strain, so the slope of the stress–strain curve will increase, which will lead to the hardening of silicone foam, while stress–relaxation leads to the reduction of actual stress, which will lead to the softening of silicone foam. This means that when stress–relaxation is dominant, the material will soften, while when the compression set is dominant, the material will harden. The change of the stress–strain curve also proves the correctness of the above conclusion.

From the stress–strain curves of the compressed silicone rubber foam before and after aging, as shown in Figure 3 and Figure 4, it can be observed that the change of the stress–strain response showed non-monotonicity with the aging time. In the early stage (aging for 0–48 h), the compression set and stress–relaxation changed rapidly, and their competition led to the swing change of the stress–strain curve. For example, at 0–8 h, a rapid change in stress–relaxation dominated the aging process leading to softening of the silicone foam, while at 8–16 h, the predominance of the compression set caused the silicone foam to harden. Furthermore, the compression set and stress–relaxation contributions were similar at 16–48 h, resulting in a substantially unchanged shape of the stress–strain curves. With the deepening of the aging degree (in this work, the aging time is greater than 48 h), the compression set and stress–relaxation changes tended to be stable. At this time, the stress–strain curve of the silicone foam continued to soften, which indicated that stress–relaxation dominates at this stage. Note that whether the compression set and stress–relaxation are dominant is not simply dependent on the magnitude of their variation (relative to the unaged specimen), and the complex functional relationship behind them needs to be studied further. This finding is different from the monotonic hardening or softening of silicone foam caused by aging described in the previous literature [12,19]. The possible reason is that the sampling interval is too long for the aging of silicon foam in previous studies, therefore, they did not observe such a phenomenon that the stress-strain curve softened first and then stiffened (or stiffened first and then softened) in the early stage of aging.

### 3.2. Ogden Hyper-Foam Model

To further analyze the variation law of the stress–strain response, a second-order Ogden hyper-foam model of silicone foam was established in this study. This model was refined by Hill [26] and Storakers [27] based on the classical Ogden model [28]. Additionally, previous studies show that a second-order Ogden hyper-foam model (i.e., N = 2) fits the experimental data well [19,21,29]. The Ogden hyper-foam model is expressed as
(7)$$U\left(\lambda_{1},\lambda_{2},\lambda_{3}\right)=\overset{2}{\underset{i=1}{\sum }}\frac{2\mu_{i}}{\alpha_{i}^{2}}\left\{\lambda_{1}^{\alpha_{i}}+\lambda_{2}^{\alpha_{i}}+\lambda_{3}^{\alpha_{i}}-3+\frac{1}{\beta_{i}}\left[{\left(\lambda_{1}\lambda_{2}\lambda_{3}\right)}^{-\alpha_{i}\beta_{i}}-1\right]\right\},\;$$
where $\lambda_{1}$, $\lambda_{2}$, and $\lambda_{3}$ are the principal elongation; $\alpha_{i}$ is the parameters related to the degree of hyper-elasticity of the material; $\mu_{i}$ and $\beta_{i}$ are parameters related to the initial shear modulus $\mu_{0}$, initial bulk modulus $K_{0}$, and Poisson’s ratio of materials $\nu_{i}$,


where

$$\mu_{0}=\sum_{i=1}^{2}\mu_{i},\;K_{0}=\overset{2}{\underset{i=1}{\sum }}2\mu_{i}\left(\frac{1}{3}+\beta_{i}\right),\;v_{i}=\frac{\beta_{i}}{1+2\beta_{i}}.\;$$



To obtain the expression of the stress–strain equation, the above equation was analyzed. For uniaxial compression tests, materials only bear load in one direction and free deformation in other directions. The loading stretch was determined as $\lambda_{1}=\lambda_{L}$, and the transverse stretches were chosen to be *λ*_2_ and *λ*_3_. Considering the isotropic deformation of silicone rubber foam, we observed $\lambda_{2}=\lambda_{3}$. Therefore, the equation can be derived as
(8)$$U\left(\lambda_{1},\lambda_{2}\right)=\overset{2}{\underset{i=1}{\sum }}\frac{2\mu_{i}}{\alpha_{i}^{2}}\left\{\lambda_{L}^{\alpha_{i}}+2\lambda_{2}^{\alpha_{i}}-3+\frac{1}{\beta_{i}}\left[{\left(\lambda_{L}\lambda_{2}{}^{2}\right)}^{-\alpha_{i}\beta_{i}}-1\right]\right\}.\;$$

The stress of the material in the loading direction can be obtained by deriving the strain in this direction from Equation (8).
(9)$$\sigma_{x}=\frac{\partial U}{\partial \varepsilon_{x}}=\frac{\partial U}{\partial \lambda_{x}}\cdot \frac{\partial \lambda_{x}}{\partial \varepsilon_{x}}.\;$$

According to the relationship between principal elongation and engineering strain, we determined that $\lambda_{x}=1+\varepsilon_{x}$.

The stress in the loading direction can be obtained by the partial derivative of the strain energy equation.
(10)$$\begin{array}{cc} \sigma_{L} & =\frac{\partial U}{\partial \varepsilon_{L}}=\frac{\partial U}{\partial \lambda_{L}}\frac{\partial \lambda_{L}}{\partial \varepsilon_{L}}=\frac{\partial U}{\partial \lambda_{L}}\\ & =\frac{2}{\lambda_{L}}\sum_{i=1}^{2}\frac{\mu_{i}}{\alpha_{i}}\left[\lambda_{L}^{\alpha_{i}}-{\left(\lambda_{L}\lambda_{2}^{2}\right)}^{-\alpha_{i}\beta_{i}}\right] \end{array}$$

By substituting $\lambda_{L}=1+\varepsilon_{L}$ into Equation (4), the stress–strain equation was obtained.
(11)$$\sigma_{L}=\frac{2}{1+\varepsilon_{L}}\overset{2}{\underset{i=1}{\sum }}\frac{\mu_{i}}{\alpha_{i}}\left\{{\left(1+\varepsilon_{L}\right)}^{\alpha_{i}}-{\left[\left(1+\varepsilon_{L}\right){\left(1-v_{i}\varepsilon_{L}\right)}^{2}\right]}^{-\alpha_{i}\beta_{i}}\right\},\;$$
where $\sigma_{L}$ is the stress of material; $\varepsilon_{L}$ is the strain of material.

In our work, the compression level was limited to 40%. In this condition, there was almost no lateral bulging for the silicone rubber foams. Therefore, the Poisson’s ratio of the silicone foam was considered to be 0 [30]. According to the relationship between parameter β and Poisson’s ratio ν, the parameters $\beta_{i}$ are taken as 0.

By substituting $\beta_{i}=0$ into Equation (11), Equation (12) can be obtained.
(12)$$\sigma_{L}=\frac{2}{1+\varepsilon_{L}}\overset{2}{\underset{i=1}{\sum }}\frac{\mu_{i}}{\alpha_{i}}\left[{\left(1+\varepsilon_{L}\right)}^{\alpha_{i}}-1\right].\;$$

The above equation is taken as the constitutive model of silicone foam material, and the parameters *μ* and *α* in the equation can be obtained by fitting the experimental data.

### 3.3. Various of Model Parameters

By fitting the stress–strain curves of silicone foam with different aging degrees, the corresponding second-order Oden constitutive model was established, and the model parameters are shown in Table 1, and the variation of parameters with time is shown in Figure 5. The results show that the four model parameters strongly depend on the aging time. The model parameters $\mu_{1}$ and $\alpha_{1}$ decrease gradually with increasing aging time, and they have the same trend of change. Parameter $\alpha_{2}$ is the largest of all parameters. Except for the initial stage, it also decreases with the extension of aging time. At the same time, the model parameter $\mu_{2}$ decreases first and then increases, which is almost consistent with the change law of the stress–strain curve. When the stress–strain curve becomes soft, parameter $\mu_{2}$ decreases, and when the stress–strain curve becomes hard, parameter $\mu_{2}$ increases. In addition, the change in the model parameters can reflect the change in the stress–strain curve. When aging for 0–16 h, the stress–strain curve first softens and then hardens. At this time, the sum of $\mu_{1}$ and $\mu_{2}$ (initial elastic modulus) first decreases and then increases, and the parameters $\alpha_{1}$ and $\alpha_{2}$ related to the degree of hyperelasticity of the material rapidly change. When aging for 16–24 h, the sum of $\mu_{1}$ and $\mu_{2}$ decreases rapidly, and the parameters $\alpha_{1}$ and $\alpha_{2}$ decrease rapidly, showing that the shape of the stress–strain curve of aging 16 h and 32 h is basically unchanged. When aging for 24–48 h, the stress–strain curve and the corresponding model parameters have small changes. Furthermore, When the aging time is greater than 48 h, the stress–strain curve changes monotonically, resulting in a monotonous change in the corresponding model parameters, which is consistent with Lou’s [19] conclusion.

Furthermore, the change of the stress–strain curve of silicone foam is related to stress–relaxation and the compression set. When the stress–strain curve changes monotonously, it means that the stress–relaxation or compression set is dominant. Previous studies have shown that in this case (aging time is greater than 48 h), the values of the model parameters vary exponentially with the increasing aging time [19], which is consistent with the change rule of stress–relaxation or the compression set. Based on the experimental observations, the following exponential functions are proposed as
(13)$$\mu \left(t\right)=a+be^{-k_{u}t},\;$$
(14)$$\alpha \left(t\right)=a+be^{-k_{\alpha }t},\;$$
where *a*, *b*, *c*, and *d* are constants. The terms *k_μ_* and *k_α_* are the kinetic rate parameters related to the temperature. The parameters in Equations (13) and (14) can be obtained by adopting the nonlinear least-square method; the fit curves are shown in Figure 6.

The fitting results indicated that Equations (13) and (14) fit the aging model parameters data well. The resulting model parameters were obtained as shown in Equation (15). The results show that although the stress–strain response of silicone foam changes and is complex in the aging process, which first softens, then hardens, and finally softens, the changing trend of the stress–strain curve can be reflected by the change of Oden model parameters. With the deepening of aging, the chain breaking or cross-linking of silicone foam rubber polymer is dominant. At this time, it shows monotonic softening or hardening. Additionally, the parameter of the Oden constitutive model of silicone foam is the exponential function of aging time.
(15)$$\left\{\begin{array}{l} \mu_{1}=-0.1463+0.5473e^{-0.0007215t}\\ \alpha_{1}=0.07091+0.1299e^{-0.00163t}\\ \mu_{2}=0.03652+0.01499e^{-0.01516t}\\ \alpha_{2}=12.68+-0.4364e^{0.005234t} \end{array}\right..\;$$

## 4. Conclusions

In this paper, through accelerated aging tests, the mechanical properties of silicone rubber foams were studied under the coupling of temperature and pressure, and the compression set, stress–relaxation, and stress–strain curves of silicone foams after aging for different times were measured. Additionally, the degradation trajectories of compression set and stress–relaxation were obtained according to the experimental data. The experimental data show that the compression set has an exponential relationship with aging time, and stress–relaxation has a logarithmic relationship with aging time. In addition, our study found some different conclusions compared to existing studies; that is, with the deepening of aging time, the change of stress–strain response shows non-monotonicity, which first softens, then hardens, and finally softens. During aging, the non-monotonic change in the stress–strain response is related to the compression set and stress–relaxation. Specifically, when the compression set dominates, the silicone foam becomes stiff, while when stress–relaxation dominates, the silicone foam becomes soft. In the early stage (aging for 0–48 h), the compression set and stress–relaxation change rapidly, and their competition leads to the swing change of the stress–strain curve. With the deepening of the aging degree (aging than 48 h), the compression set and stress–relaxation changes tend to be stable. At this time, the stress–strain curve of the silicone foam continues to soften, which indicates that stress–relaxation dominates at this stage.

To further analyze the change law of the stress–strain response during the aging process, a two-order hyper-foam model was established, and the changes in model parameters were studied. The results show that the change in the model parameters can reflect the change in the stress–strain curve. During aging at 0–24 h, the stress–strain curve changes rapidly, resulting in rapid changes in the corresponding parameters. At 24–48 h of aging, the shape of the stress–strain curve is basically unchanged, so the corresponding parameters are also stable. After aging for more than 48 h, the stress and strain change monotonically, and the model parameters also change monotonically at this time, which changes exponentially with the increase of aging time.

## Figures and Tables

**Figure 1 polymers-14-03606-f001:**
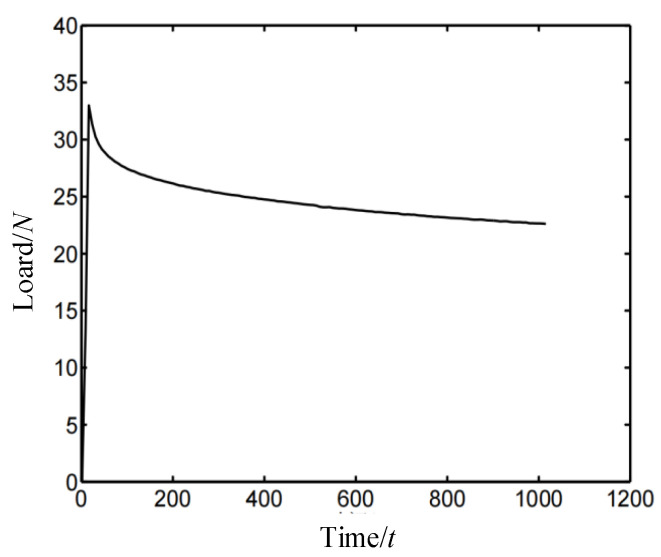
Stress–relaxation curve of silicone foam.

**Figure 2 polymers-14-03606-f002:**
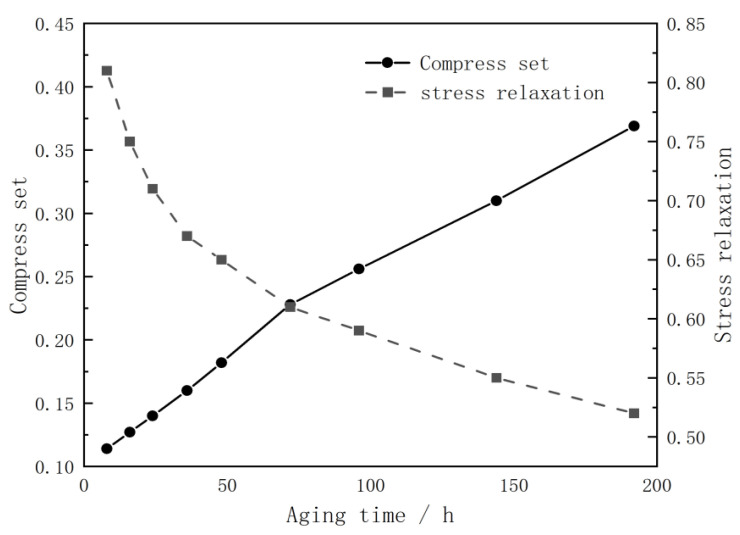
Compress set and stress–relaxation curves for aged silicone rubber foams under thermal conditions of 125 °C.

**Figure 3 polymers-14-03606-f003:**
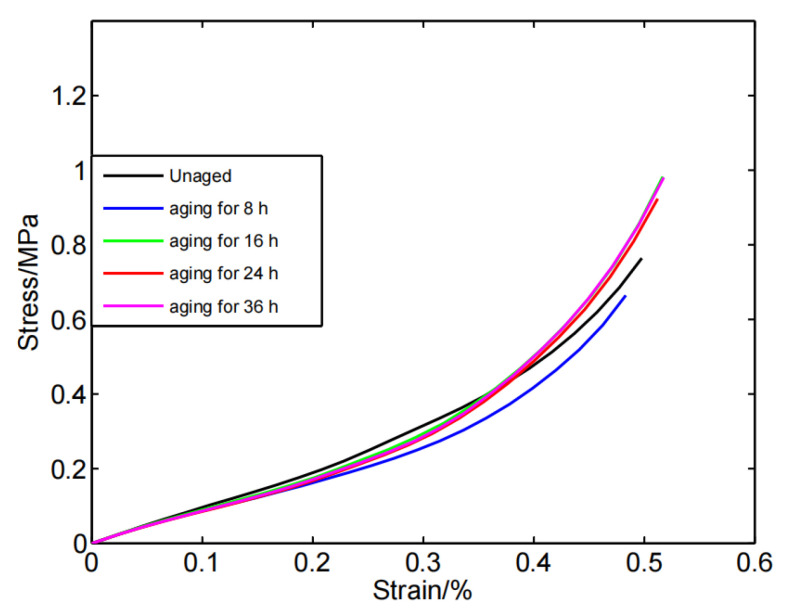
The stress–strain curves for silicone foam aged in the early stage.

**Figure 4 polymers-14-03606-f004:**
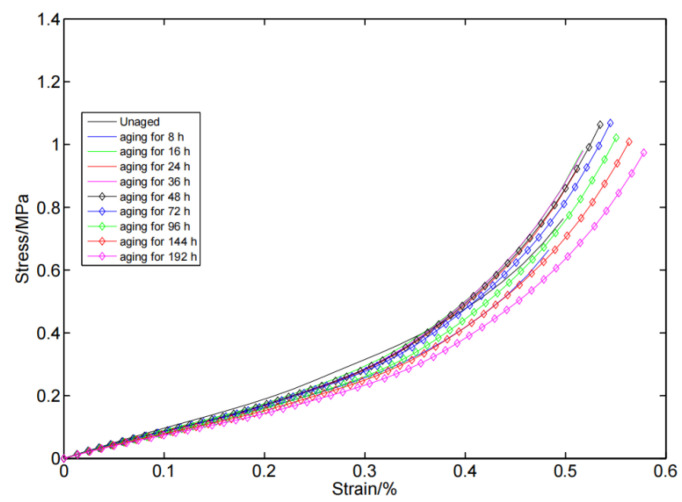
The stress–strain curves for silicone foam aged for different times.

**Figure 5 polymers-14-03606-f005:**
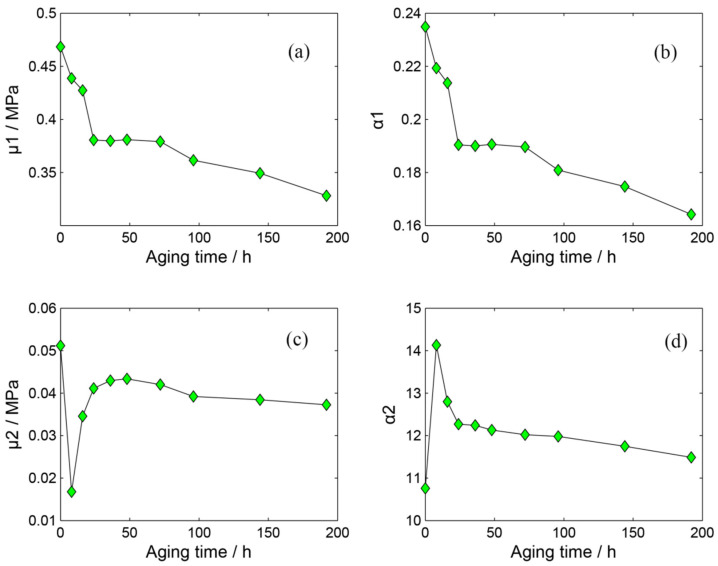
Variation of hyper-foam model parameters with time at different degrees of aging: (**a**) $\mu_{1}$; (**b**) $\alpha_{1}$; (**c**) $\mu_{2}$; (**d**) $\alpha_{2}$.

**Figure 6 polymers-14-03606-f006:**
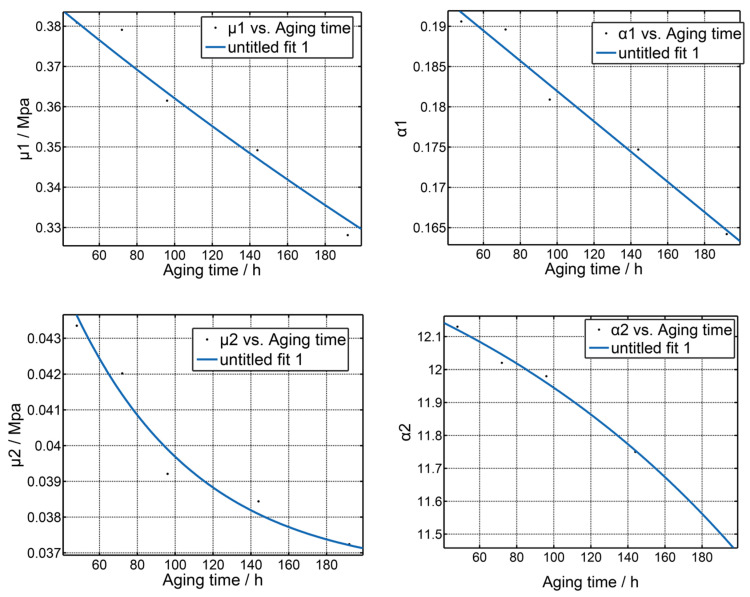
The fitting results of the hyper-foam model parameters as functions of aging time: $\mu_{1}$, $\alpha_{1}$, $\mu_{2}$, and $\alpha_{2}$.

**Table 1 polymers-14-03606-t001:** Constitutive model parameters at various temperatures.

Aging Time/h	$\mu_{1}$	$\alpha_{1}$	$\mu_{2}$	$\alpha_{2}$	$\mu_{1}$ $+\mu_{2}$
0	0.4684	0.2349	0.05117	10.76	0.5196
8	0.4387	0.2193	0.01676	14.13	0.4555
16	0.4272	0.2137	0.03456	12.8	0.4618
24	0.3806	0.1904	0.04111	12.27	0.4217
36	0.3797	0.19	0.04296	12.24	0.4227
48	0.3809	0.1906	0.04335	12.13	0.4243
72	0.3791	0.1896	0.04202	12.02	0.4211
96	0.3615	0.1809	0.03921	11.98	0.4007
144	0.3492	0.1747	0.03844	11.75	0.3876
192	0.3281	0.1642	0.03724	11.49	0.3653

## Data Availability

The data presented in this study are available on request from the corresponding author.

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
