# Peer review of "The Variety of the Stress–strain Response of Silicone Foam after Aging"

_polymers, 2022, doi:10.3390/polym14173606_

Round 1
Reviewer 1 Report
In this work, the authors study the variety of stress-strain responses for silicon foam in aging, the alteration of compress set, stress relaxation, and strain-stress response under long-term compressive strain was examined by accelerated aging tests, and the Hyperfoam constitutive model was tailored rendering to the experimental data. The results exhibit that the compressed set and stress relaxation are exponential functions of time. While, the stress-strain curves do not exchange monotonically with increasing time, which first softens, then hardens, and finally softens. In the stage of monotonic change of stress-strain curve, the variation of constitutive model parameters shows a regular in aging that the model parameters vary exponentially with the aging time increase. Therefore, I recommend this manuscript for publication in molecules after major revision.
Introduction:
1- the abstract and conclusion should be revised carefully and remove the inserted references from the conclusion part.
2- The text should be revised and omit any minor English mistakes.
3- Insert the website addresses of the companies in the experimental part
4- Is there any availability to detect the influences of the studied properties using TEM, SEM and …..etc?
Author Response
Dear reviewer:
Thank you for your valuable comments. Your comments are very helpful for us to improve the quality of the paper. For your comments:
1. the abstract and conclusion should be revised carefully and remove the inserted references from the conclusion part.
Re: We have reorganized the Abstract and Conclusions sections and removed references in the Conclusions section.
2. The text should be revised and omit any minor English mistakes.
Re: Corrected spelling and grammatical errors in the article.
3. Insert the website addresses of the companies in the experimental part.
Re: Provided the company URL in the experimental part.
4. Is there any availability to detect the influences of the studied properties using TEM, SEM and …..etc?
Re: Thank you for your valuable opinions. It will be helpful to better understand the changes of mechanical properties of silicon foam by using TEM and SEM to observe the structural changes of silicon foam before and after aging. However, we are sorry that our unit cannot provide relevant instruments, so the corresponding test cannot be carried out.
All corrections were made in revision mode, and the corrections were highlighted.
Kind regards

Reviewer 2 Report
- Manuscript has some typos, revise carefully and correct it
- 2.1. Include information about purification materials or include sentence “used as received” and include the purchased information for all materials
- Page 2, lines 82 and 83 change “8 h, 16 h, 24 h, 36 h, 48 h, 72 h, 96 h 144h 82 and 192h” for “8, 16, 24, 36, 48, 72, 96, 144, and 192 h”
- Improve quality for Figures 2, 5 and 6 at minimal 300 dpi, because in the present form have very poor resolution.
- Manuscript has two cited forms, see line 224 (references 5 and 12)
- Include important results in the conclusion part
- Manuscript has some interesting results but doesn’t have discussion, include it for all Figures
- Manuscript has 2 references from 2021 and 2 from 2022, include more recently references
- Include correction with different color please for easy revision
Author Response
Dear reviewer:
Thank you for your valuable comments. Your comments are very helpful for us to improve the quality of the paper. For your comments:
- Manuscript has some typos, revise carefully and correct it.
Re: Corrected spelling and grammatical errors in the article
- 1. Include information about purification materials or include sentence “used as received” and include the purchased information for all materials.
Re: Modifications have been made based on your valuable comments.
- Page 2, lines 82 and 83 change “8 h, 16 h, 24 h, 36 h, 48 h, 72 h, 96 h 144h 82 and 192h” for “8, 16, 24, 36, 48, 72, 96, 144, and 192 h”.
Re: Modifications have been made based on your valuable comments
- Improve quality for Figures 2, 5 and 6 at minimal 300 dpi, because in the present form have very poor resolution.
Re: Replaced Figures 2, 5 and 6 with a clearer picture.
- Manuscript has two cited forms, see line 224 (references 5 and 12).
Re: The reference format was carefully checked and unified.
- Include important results in the conclusion part.
Re: Rewrite the conclusion part and include important results.
- Manuscript has some interesting results but doesn’t have discussion, include it for all Figures.
Re: The results obtained are discussed in more detail.
- Manuscript has 2 references from 2021 and 2 from 2022, include more recently references.
Re: More relevant references in recent years are added
- Include correction with different color please for easy revision
Re: All corrections were made in revision mode, and the corrections were highlighted.
Kind regards
Round 2
Reviewer 1 Report
The author responded to all comments significantly and the paper in the final version is suitable for publication in Polymer journal